# Watt-Level, Narrow-Linewidth, Tunable Green Semiconductor Laser with External-Cavity Synchronous-Locking Technique

**DOI:** 10.3390/s25216758

**Published:** 2025-11-05

**Authors:** Chunna Feng, Bangze Zeng, Jinhai Zou, Qiujun Ruan, Zhengqian Luo

**Affiliations:** 1Fujian Key Laboratory of Ultrafast Laser Technology and Applications, Xiamen University, Xiamen 361005, China; fcnnacho@163.com (C.F.); jinhaizou24@163.com (J.Z.); 15906067687@163.com (Q.R.); 2School of Information and Communication Engineering, University of Electronic Science and Technology of China, Chengdu 611731, China; zengbangze@163.com

**Keywords:** green laser, external-cavity semiconductor laser, synchronous-locking

## Abstract

External-cavity GaN semiconductor lasers at blue wavelengths enable narrow-linewidth and high-power output but is difficult at >500 nm green wavelengths due to the so-called ‘green gap’. In this Letter, we demonstrate a watt-level, narrow-linewidth, tunable green semiconductor laser based on external-cavity synchronous-locking technique. The laser consists of two green edge-emitting laser diodes (LDs), beam-shaping devices and a visible-wavelength diffraction grating. Because the two green (∼518 nm) LDs have similar spectral and lasing characteristics and are adjacently parallel in spatial mode, synchronous locking of both beam can readily generated with the help of diffraction grating. Namely, the two green LDs are locked at the same wavelength and the 3dB–linewidth is sharply narrowed from 4 nm to 0.06 nm. The locked wavelength can be tuned from 512.2 to 520.2 nm. The maximum output power reaches 1.53 W at 518 nm with a 3dB–linewidth of 0.15 nm. This is, for the first time, to the best of our knowledge, an external-cavity synchronous-locking green semiconductor laser with watt-level output power.

## 1. Introduction

High-power narrow-linewidth green semiconductor lasers have attracted much attention in some applications, such as laser holographic display [1,2,3], biomedicine [4,5,6], underwater communications and detection [7,8] and pump sources of Ti: sapphire femtosecond lasers [9,10,11]. It is well known that several techniques can be used to generate high-power narrow-linewidth semiconductor lasers, including distributed feedback (DFB) [12], distributed Bragg reflection (DBR) [13] and external-cavity feedback [14,15]. Recently, DFB and DBR techniques have been adopted to achieve narrow-linewidth GaN laser diodes (LDs) at blue wavelengths [16,17], but DFB or DBR green LD is very challenging. In contrast, external-cavity semiconductor lasers (ECDLs) can not only obtain narrow-linewidth laser but also have the advantages of simple structure and easy tunning [18], which is an ideal solution to realize green high-power narrow-linewidth semiconductor laser.

In recent years, a few advances in green ECDLs have been reported [19]. In 2016, Jensen et al. reported a 515 nm green ECDL with output power up to 480 mW, a tuning range of 9.2 nm and a linewidth of 8 pm [20]. Then, the linewidth was further narrowed to 10 MHz at 518 nm, realizing the output power of 40 mW and the tuning range of 8 nm [21]. These efforts have been paid to enhance the characteristics of green ECDLs such as spectral linewidth and tuning range, however, the green ECDLs still suffer from the low output power, limited to some practical applications. The low output power mainly originates from (1) the limited power (∼1 W) of single green LD due to the well-known “green gap” [22], and (2) the simple scheme using only one diode laser. In order to obtain the high-power narrow-linewidth ECDLs, synchronous-locking technique of multiple LDs has been verified to be an effective method in the blue and near-infrared spectral range [23,24,25,26,27]. Nevertheless, ECDL with multiple-LD scheme has had no progress in green wavelength. Therefore, it is worth exploring high-power narrow-linewidth green ECDLs with synchronous-locking of multiple green LDs.

In this Letter, we experimentally demonstrated a watt-level narrow-linewidth tunable green ECDL by proposing external-cavity synchronous-locking of two green GaN LDs. The green wavelength can be tunable in the range of 512.2–520.2 nm, and the 3dB–linewidth is as narrow as 0.06 nm. The maximum output power of 1.53 W is obtained at 518 nm. To the best of our knowledge, this is the first demonstration of watt-level green ECDL using the synchronous-locking technique.

## 2. Materials and Methods

The experimental setup of the tunable green ECDL with beam shaping is shown in Figure 1a. The two 518 nm edge-emitting LDs are commercially available with maximum output power of 1.65 W (Nichia, NUGM06T), having an estimated stripe width of 20 µm. Their output beams are shaped through fast axis collimations (FACs) and slow axis collimations (SACs), with focal length of 0.3 and 0.8 mm. After beam shaping, the two beams possess small divergence full–angles of 1.35 and 1.13 mrad and spot size of 4.4 and 4.0 mm in the directions of fast and slow axis, respectively. Spatial beam-combining technology was then used to the two LDs for obtaining higher green-light output power. As shown in Figure 1a, the combined beam profile was detected by a visible-wavelength CCD. The two beams can be combined to be very close in space and have good beam quality, which play significant roles to realize synchronous-locking of the two LDs. To ensure the stability of system, the two LDs are placed on a thermoelectric cooler with a constant temperature of 23 °C. The diffraction grating operating in the visible spectral region acts as an external-cavity feedback element. The grating grooves are parallel to the slow axis of the LDs. The 0th–order diffraction beam serves as the output while the 1st–order diffraction beam is feedback into the LDs for the synchronous-locking technique. The cavity length is around 250 mm and the lasing wavelength of ECDL is tuned by rotating the grating. Figure 1b depicted the operation principle of the synchronous-locking technique. Both the green LDs have similar fluorescence and freely running spectra. In principle, once the grating external-cavity feedback is formed, one of the two LDs will be firstly locked by self-matching the strong fluorescence spectrum with the grating feedback profile, and then the wavelength-locked LD light is partially injected into the other LD to initiate their synchronous locking. Finally, narrow-linewidth and high-power green laser by the external-cavity synchronous-locked LDs are obtained. Compared with the limited power of ECDL using single green LD, two or multiple LDs by such external-cavity synchronous-locking technique can significantly enlarge the total green-light output power with the high performance of narrow linewidth and wide spectral tunability.

## 3. Results and Discussion

In order to optimize the output performance of the green ECDL, three gratings with different grooves and diffraction efficiency are employed as the external-cavity feedback element. The 0th–order and 1st–order diffraction efficiencies of these gratings at the Littrow angle are compared in Table 1. Specifically, the gratings A and B have the same groove density (1800 lines/mm) with different blazed wavelengths of 500 nm and 250 nm. The gratings B and C show a same blazed wavelength of 250 nm with different groove density of 1800 and 2400 lines/mm. The 1st–order diffraction efficiency of the three gratings around 520 nm is 40.9%, 15.5% and 5.0%, respectively. Considering the insertion loss of the gratings, the output coupling (i.e., the 0th order efficiency) of the green ECDL is 48.1%, 66.6% and 84.6%, respectively.

Subsequently, we comparatively investigated the external-cavity synchronous-locking of the green ECDL using the three gratings. Under an injection current of 800 mA, the output spectra of the green laser in freely running or external-cavity-locking regime are measured by an optical spectrum analyzer (Ando. AQ-6315E). As shown in Figure 2a, the green laser in freely−running regime operates in multiple Fabry–Perot modes with a wide 3 dB–linewidth of ∼4 nm at 518 nm center wavelength. In contrast, all the output optical spectra of the green ECDL under the three different gratings (A, B and C) can exhibit single-peak operation and narrower 3dB–linewidth of less than 0.15 nm, manifesting the occurrence of external-cavity synchronous-locking. Figure 2b compares the threshold current of the green ECDL under the three gratings in the wavelength-tunable range of ∼511 to 521 nm. Compared with the threshold current of 210 mA in the free-running regime, all the external-cavity synchronous-locking operations show lower threshold currents between 135 and 188 mA. Although the threshold currents’ difference among the three gratings is not large, the grating A always gives the lowest threshold current due to its highest 1st–order diffraction efficiency of 40.9%. Moreover, when the lasing wavelength is tuned in the range of 511.5–520.5 nm, the lowest threshold current appears around 518 nm, because the highest gain of the used green LDs locates at 518 nm. Figure 2c shows the P-I characteristics of the green laser operating in free-running or external-cavity synchronous-locking by the three different gratings, respectively. All of them are fixed at the wavelength of 518 nm. The slope efficiencies for the free running and the external-cavity synchronous-locking by the gratings (A, B and C) are measured to be 1.438, 0.365, 0.595 and 1.05 W/A, respectively. Clearly, the efficiency and output power of the green ECDL with the grating C is significantly higher than that with the gratings A and B. Therefore, in our next experiments, we choose the grating C as a feedback element for synchronous-locking of the green ECDL.

Figure 3 summarized the typical characteristics of the green ECDL based on the grating C. When the lasing wavelength was locked at 518.7 nm, the optical spectral evolution of the green ECDL as the injection current is shown in Figure 3a. It should be noticed that, although two LDs were utilized, the center wavelength remained unchanged and the narrow-linewidth lasing could be always observed from the threshold current to 1.3 A, indicating that the external-cavity synchronous-locking operation is very stable. Figure 3b gives the spectral linewidth of the green ECDL under different injection currents. The linewidth slightly broadened from 0.06 to 0.15 nm with the increase of output power from 30 to 1539 mW (corresponding to the injection current from 155 to 1400 mA). The P-I characteristic of the ECDL is depicted in Figure 3c. The threshold current is approximately 150 mA, and the slope efficiency is around 1.23 W/A. Thanks to the synchronous-locking technique of two green LDs, 1.53 W output power was finally obtained with narrow linewidth under a maximum injection current of 1400 mA. As plotted in Figure 3d, we evaluated the power stability of the green ECDL operating at 800 mA during a 30 min test. The results exhibit a power fluctuation of only 0.1%, indicating excellent power stability of the ECDL.

The wavelength tunability of the green ECDL was also studied by rotating the Littrow angle of the grating. In our experiment, we found that the widest wavelength-tuning range took place near the threshold current (∼150 mA) of the ECDL, but the output power was too low (only a few milliwatts). To easily measure the tunable spectra, we increased the injection current to 210 mA and measured the typical tunable spectra of the ECDL. As plotted in Figure 4a, the wavelength-tuning range can be as wide as 8.0 nm from 512.2 nm to 520.2 nm. The amplified spontaneous emission (ASE) was well suppressed in the whole tuning range, especially when the ECDL operated near 518 nm (i.e., the highest gain region of the green LDs). Although the ASE became observable when the ECDL operated at the edges of the wavelength-tuning range, the spectral signal-to-noise ratio is still more than 25 dB. Figure 4b illustrates the wavelength-tuning range and the output power under different injection currents from 210 mA to 1400 mA. One can see that the tuning range gradually reduced from 8.0 nm to 2.4 nm as the injection current increased to 1000 mA. The decrease in tuning range is mainly determined by the balance between the gain profile and the lasing competition of the inner F-P cavity. Interestingly, when we increased the injection current to 1400 mA, the tuning range reversely broadened to 2.9 nm, which could be attributed to the combination of the thermal effect and band-filling effect [28]. As the competitive nature between the ECDL resonance and the inner F-P cavity resonance becomes serious, it becomes challenging to completely suppress the F-P lasing, limiting the wavelength-tuning range. What is more, the corresponding output power increases from several milliwatts to 1539 mW with the increasing of injection current. Although the output power reduces slightly when the ECDL is tuned away from the center of the gain profile, obviously the power difference in the whole wavelength-tuning range is less than 10% under each fixed current.

## 4. Conclusions

In conclusion, a watt-level, narrow-linewidth tunable green ECDL based on the synchronous–locking technique has been demonstrated. After the two 518 nm LDs with the similar gain profile were beam–shaped and spatially beam–combined, their synchronous locking by a visible external–cavity grating was successfully realized. The output performance (power, wavelength–tuning range, etc.) of the green ECDL was comprehensively investigated by optimizing the gratings, injection current and beam shaping. A total of 1.5 W output power at 518 nm was achieved and the 3dB–linewidth was as narrow as 0.15 nm; the output power is three times that of previous reports. Moreover, the green ECDL enables a wide wavelength-tuning range of 8.0 nm from 512.2 to 520.2 nm. Such technique may further upgrade by synchronous-locking multiple LDs to scale up narrow-linewidth green-light output power to tens of watts or even higher level for applications in underwater communications and detection, bio-photonics and holographic display.

## Figures and Tables

**Figure 1 sensors-25-06758-f001:**
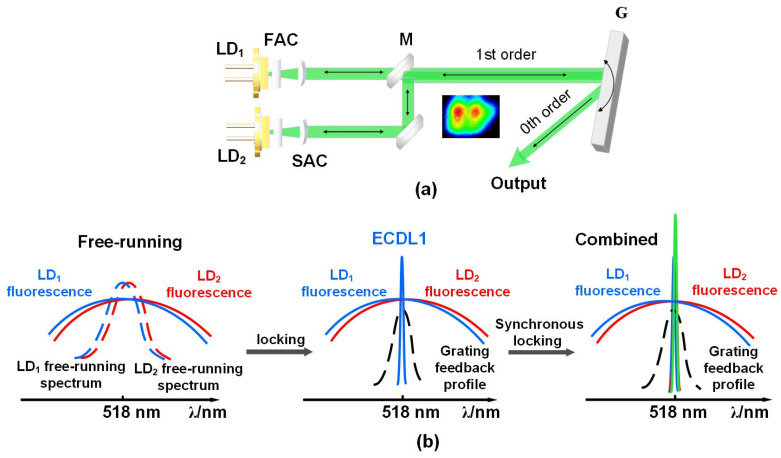
(**a**) Experimental setup of the green ECDL system. LD1 and LD2: 518 nm diode laser; FAC: fast axis collimator, SAC: slow axis collimator; M: green high reflective mirror; G: diffraction grating; (**b**) Schematic operation principle of the external-cavity synchronous-locking.

**Figure 2 sensors-25-06758-f002:**
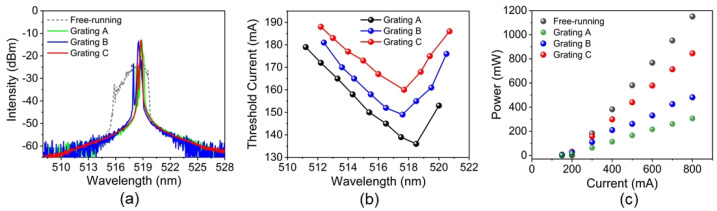
(**a**) Optical spectra of the green laser in freely running or external–cavity–locking regime under an injected current of 800 mA; (**b**) The threshold current as a function of the tuning wavelength for the ECDL with different gratings; (**c**) The output power versus the injection current for the free-running laser and the ECDL with different gratings at 518 nm.

**Figure 3 sensors-25-06758-f003:**
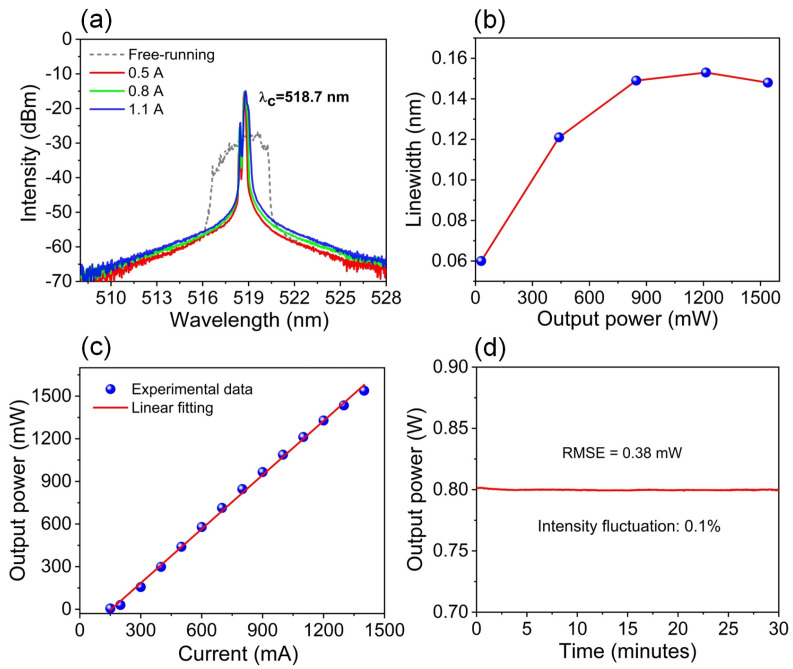
(**a**) Output optical spectra of the ECDL under different injection currents (dash: freely running green laser at 1100 mA); (**b**) Linewidth of the ECDL as a function of the output power; (**c**) Optical output power of the ECDL versus injected current of two LDs; (**d**) Power stability measurement during 30 min.

**Figure 4 sensors-25-06758-f004:**
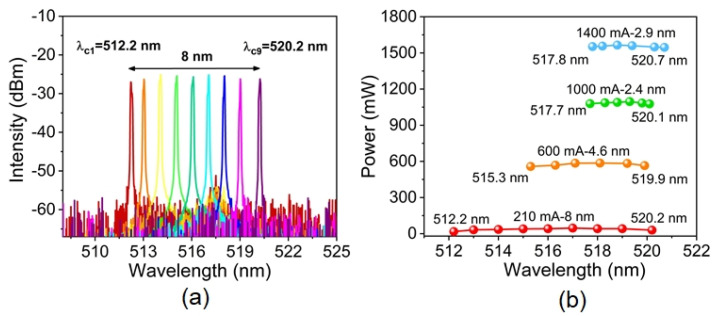
(**a**) The typical wavelength tuning of the green ECDL at the injected current of 210 mA; (**b**) The output power of the green ECDL in the tuning range under different injection currents.

**Table 1 sensors-25-06758-t001:** List of grating parameters.

Grating	Ruling Density (Lines/mm)	Blazed Wavelength (nm)	0th Order Efficiency	1st Order Diffraction Efficiency
A	1800	500	48.1%	40.9%
B	1800	250	66.6%	15.5%
C	2400	250	84.6%	5.0%

## Data Availability

Data are contained within the article.

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
