# Peer review of "Watt-Level, Narrow-Linewidth, Tunable Green Semiconductor Laser with External-Cavity Synchronous-Locking Technique"

_sensors, 2025, doi:10.3390/s25216758_

Round 1

Reviewer 1 Report

Comments and Suggestions for Authors

High-power narrow-linewidth green semiconductor lasers have broad  applications. This manuscript reports a watt-level tunable narrow-linewidth green external-cavity diode laser (ECDL), which achieves green laser output with a tuning range of 8 nm and a linewidth of 0.06 nm. The research results of this paper possess good practical value and the manuscript can be published after appropriate revisions. Specific revision suggestions are as follows:

(1) In Fig. 1(a), the beam combining of the two LDs adopts a mirror with high reflectivity for green light. If this is the case, how does the light from LD1 pass through the mirror? If the light from LD1 does not pass through the mirror, how does the author achieve the coaxial alignment of the beams from LD1 and LD2 in space? To improve the readability of the paper, the authors need to elaborate on the specific method for realizing spatial beam combining of the two LDs.

(2) Spatial beam combining of two beams usually degrades the beam quality. The authors are required to measure the far-field spot of the laser after beam combining, such as the M² factor, far-field spot size, and divergence angle.

(3) Do the two LDs have built-in optical isolators? If yes, what is the isolation degree? The isolation degree affects the stability of the external-cavity laser and the results of linewidth compression, so the authors should add a detailed discussion about the two LDs. If there are no optical isolators, how to ensure that the feedback light does not damage the LDs under such high power conditions?

(4) In Fig. 3(c), it is necessary to clarify whether the injection current values on the abscissa represent the injection current of a single LD or the total injection current of the two LDs.

(5) The abscissa of Fig. 3(b) is inconsistent with the description in its caption.

(6) Parts of Fig. 3(a) and Fig. 3(c) are occluded and cannot be fully observed.

Author Response

(Note: the reviewer’s comments are in italic and blue colors.)

We would like to express our sincere thanks to the reviewer for his/her valuable comments and suggestions. We have revised the manuscript in accordance with the reviewer’s comments and suggestions. All the changes made in the revision are underlined. Our replies to the reviewer’s comments and suggestions are as follows.

High-power narrow-linewidth green semiconductor lasers have broad  applications. This manuscript reports a watt-level tunable narrow-linewidth green external-cavity diode laser (ECDL), which achieves green laser output with a tuning range of 8 nm and a linewidth of 0.06 nm. The research results of this paper possess good practical value and the manuscript can be published after appropriate revisions. Specific revision suggestions are as follows:

  1.  In Fig. 1(a), the beam combining of the two LDs adopts a mirror with high reflectivity for green light. If this is the case, how does the light from LD1 pass through the mirror? If the light from LD1 does not pass through the mirror, how does the author achieve the coaxial alignment of the beams from LD1 and LD2 in space? To improve the readability of the paper, the authors need to elaborate on the specific method for realizing spatial beam combining of the two LDs.

Response: We appreciate the comments of the reviewer. As you say, the light from LD1 does not pass through the mirror, it is directly incident on the grating. We adopt the spatial beam combining principle. The light from LD1 does not need to be reflected by a mirror and can be directly transmitted to the grating. the light from LD2 needs to be reflected by mirrors 1 and 2 respectively, and finally coupled to the grating in the same direction. It is necessary to use fast and slow axis collimating mirrors on the LDs to achieve beam collimation and shaping.

  1. Spatial beam combining of two beams usually degrades the beam quality. The authors are required to measure the far-field spot of the laser after beam combining, such as the M² factor, far-field spot size, and. divergence angle

Response: The far-field spot of the laser after beam combining are shown in Fig. 1s. After beam shaping, the two beams divergence full-angles of 1.35 and 1.13 mrad and far-field spot size of 4.4 and 4.0 mm in the directions of fast and slow axis. The related content has been added to the 2nd paragraph, page 2 of the revised manuscript.

Fig.1s. The far-field spot of the laser after beam combining.

  1.  Do the two LDs have built-in optical isolators? If yes, what is the isolation degree? The isolation degree affects the stability of the external-cavity laser and the results of linewidth compression, so the authors should add a detailed discussion about the two LDs. If there are no optical isolators, how to ensure that the feedback light does not damage the LDs under such high power conditions?

Response: Thank you very much for pointing out the unclear points. In our experiment, there are no optical isolators of LDs. The external-cavity semiconductor laser (ECDL) achieves mode selection by utilizing an external cavity feedback element, which selectively reduces the losses for the desired wavelength through optical feedback from the external cavity. This process effectively inhibits the oscillation of other wavelengths, resulting in a multi-mode output with a narrowed linewidth. Only a weakly light feedback is needed, the 1st-order diffraction efficiency of the gratings around 520 nm is only 5.0%,  so it will not cause damage.

  1. In Fig. 3(c), it is necessary to clarify whether the injection current values on the abscissa represent the injection current of a single LD or the total injection current of the two LDs.

Response: The injection current values on the abscissa represent the injection current of the total injection current of the two LDs. The related contents have been modified in the Fig. 3(c), page 4 of the revised manuscript.

  1. The abscissa of Fig. 3(b) is inconsistent with the description in its caption.

Response: We thank the reviewer for this remark. The related contents have been modified in the Fig. 3(b), page 4 of the revised manuscript.

  1. Parts of Fig. 3(a) and Fig. 3(c) are occluded and cannot be fully observed.

Response: Thank once again the reviewer very much for the valuable comments. The relevant contents have been modified in the Fig. 3(a) and Fig. 3(c), page 4 of the revised manuscript.

Reviewer 2 Report

Comments and Suggestions for Authors

This work is actually research on tunable green semiconductor laser. Watt-level, narrow-linewidth, tunable green semiconductor laser with external-cavity synchronous-locking technique is research in detail.I suggest revise the article, it is as follows:

  1. Semiconductor laser and laser diode appear multiple times in the text and need to be unified,please check the entire paper for similar errors.
  2. In line33, diode lasers(LDs), abbreviations do not match, please check the entire paper for similar errors.
  1. In line35, external-cavity semiconductor lasers (ECDLs),abbreviations do not match, please check the entire paper for similar errors.
  2. In Figure 1, there are two optical paths. If M is a high reflection mirror for green light, for LD1, when the green light is transmitted to M, it will be reflected, and the green light emitted from behind M will be very weak. From the experimental setup, it is not feasible. Suggest the author to provide a detailed description of the coating technology specifications for M-green high reflection mirrors and how to achieve dual channel LD green light tuning through gratings
  3. In line 95, The grating line density lines/mm should be consistent with the table 1 line density grooves/mm,please check the entire paper for similar errors.
  4. In line 101, the annotation of the table should be on top of the table, please check the entire paper for similar errors.
  5. In line 114, 115, 117, threshold current, not threshold,please check the entire paper for similar errors.
  6. The last value on the horizontal axis in Figure 3 (a) , (c) is not displayed in full.
  7. In line 188,Littrow gratings?
  8. In line 70, 88, 193, green power?
  9. The latest reference cited by the author is from 2023, and no references from the past two years have been introduced. The research results have not been compared with those of the past two years. It is recommended to cite several references from the past two years for more convincing research results.

Author Response

(Note: the reviewer’s comments are in italic and blue colors.)

We would like to express our sincere thanks to the reviewer for his/her valuable comments and suggestions. We have revised the manuscript in accordance with the reviewer’s comments and suggestions. All the changes made in the revision are underlined. Our replies to the reviewer’s comments and suggestions are as follows.

This work is actually research on tunable green semiconductor laser. Watt-level, narrow-linewidth, tunable green semiconductor laser with external-cavity synchronous-locking technique is research in detail.I suggest revise the article, it is as follows:

1.Semiconductor laser and laser diode appear multiple times in the text and need to be unified,please check the entire paper for similar errors

Response: Thank you very much for pointing out our mistakes. We have modified the formatting in the new revised manuscript, in the paragraph line 46-51 in the right column of the page 2.

2.In line33, diode lasers(LDs), abbreviations do not match, please check the entire paper for similar errors.

Response: Thank you very much for pointing out our mistakes. We have modified in the new revised manuscript, in the paragraph line 15 and 33 in the page 1.

3.In line35, external-cavity semiconductor lasers (ECDLs), abbreviations do not match, please check the entire paper for similar errors.

Response: Thank you very much for pointing out our mistakes. We have modified in the paragraph line 35 in the page 1.

4.In Figure 1, there are two optical paths. If M is a high reflection mirror for green light, for LD1, when the green light is transmitted to M, it will be reflected, and the green light emitted from behind M will be very weak. From the experimental setup, it is not feasible. Suggest the author to provide a detailed description of the coating technology specifications for M-green high reflection mirrors and how to achieve dual channel LD green light tuning through gratings

Response: We admire the reviewer for this valuable comment.

Q1: The light from LD1 does not pass through the mirror, it is directly incident on the grating. We adopt the spatial beam combining principle. The light from LD1 does not need to be reflected by a mirror and can be directly transmitted to the grating. The light from LD2 needs to be reflected by mirrors 1 and 2 respectively, and finally coupled to the grating in the same direction. It is necessary to use fast and slow axis collimating mirrors on the LDs to achieve beam collimation and shaping.

Q2: Using an ion beam-assisted deposition system and a specific electric field manipulation of film layers. Firstly, Ta2O5 (n = 2.09) and SiO2 (n = 1.46) are selected as high and low refractive index materials due to the high intrinsic damage threshold of this material couple. Secondly, the thickness of each film layer needs to be optimized to reduce the Electric Field Intensity (EFI) of the standing wave in the layers and shift the EFI peak into the SiO2 layer with better damage resistance ability. Finally, by depositing the alternately distributed Ta2O5 and SiO2 dielectric film layers on the mirrors using an ion beam-assisted deposition system (DJ-800, Jindian Vacuum), successfully fabricate coating in the green light bands.

Q3: In principle, once the grating external-cavity feedback is formed, one of the two LDs will be firstly locked by self-matching the strong fluorescence spectrum with the grating feedback profile, and then the wavelength-locked LD light is partially injected into the other LD to initiate their synchronous locking. Finally, narrow-linewidth and high-power green laser by the external-cavity synchronous-locked LDs is obtained.

5.In line 95, The grating line density lines/mm should be consistent with the table 1 line density grooves/mm, please check the entire paper for similar errors.

Response: Thank you very much for pointing out our mistakes. We have modified in the new revised manuscript, in the table 1, in the page 3.

6.In line 101, the annotation of the table should be on top of the table, please check the entire paper for similar errors.

Response: Thank you very much for pointing out our mistakes. We have modified in the new revised manuscript, in the paragraph line 101 in the page 2.

7.In line 114, 115, 117, threshold current, not threshold, please check the entire paper for similar errors.

Response: Thank you very much for pointing out our mistakes. We have modified in the new revised manuscript, in the paragraph line 114, 115, 117, in the page 3.

8.The last value on the horizontal axis in Figure 3 (a) , (c) is not displayed in full.

Response: Thank you very much for pointing out our mistakes. We have modified in the new revised manuscript, in Figure 3 (a), (c) in the page 4.

9.In line 188,Littrow gratings?

Response: Thank you very much for pointing out our mistakes. We have modified in the new revised manuscript, in the paragraph line 188, in the page 6.

10.In line 70, 88, 193, green power?

Response: Thank you very much for pointing out our mistakes. What we want to convey is the output power of the green light, and we have changed "A" to "B".We have modified “green power” to “green-light output power” in the new revised manuscript, in the paragraph line 70, 88, 193.

11.The latest reference cited by the author is from 2023, and no references from the past two years have been introduced. The research results have not been compared with those of the past two years. It is recommended to cite several references from the past two years for more convincing research results.

Response: We appreciate the comments of the reviewer. The relevant research literature in recent years has already covered this topic. There have no new breakthroughs in green light in the past two years, so our manuscript has certain value.

Reviewer 3 Report

Comments and Suggestions for Authors

This manuscript presents a watt-level, narrow-linewidth, tunable green external-cavity semiconductor laser (ECDL) with external-cavity synchronous-locking technique. The ECDL is composed of two green edge-emitting diode lasers (LDs), beam shaping devices and a diffraction grating. One of the LDs is locked by self-matching the spectrum of the diffraction grating, and then part of the locked light is injected into the other LD to complete synchronous locking.

The following is recommended to address properly before it can be recommended to be published.

  1. Why develop Watt-level green lasers, and what are the primary application requirements they address?
  2. In the introduction, the author mentions that “the DFB or DBR green LD is very challenging.” In what specific aspects does the challenge of the DFB or DBR green LD manifest?

For the description of experimental details and measurement results, the following aspects need to be explained.

  1. Is the beam profile in Fig. 1(a) measured before or after the diffraction grating?
  2. In Fig. 2, what is the operating state of the green laser during free-running? During free-running, does the feedback light from the diffraction grating return to the LD? In Fig. 2 (c), the output power of the laser free-running mode is higher than that of the other three diffraction grating states. What is the reason for the higher output power of the laser free-running mode?
  3. In the testing of the three diffraction gratings, are the tilt angles of the diffraction gratings the same, and is the laser incidence angle consistent for the different diffraction gratings?
  4. What’s the laser linewidth when the first LD is locked by the grating feedback? What method was used to measure the laser linewidth in Figure 3(b)? What is the current order of magnitude for the linewidth level of green lasers?

For some language expressions, authors are asked to confirm.

  1. The paragraphs in the text are too long. It would be better to break them into appropriate sections.
  2. Place the table title from Table 1 above the table.
  3. Place the image and caption for Figure 4 on a single page.

Author Response

(Note: the reviewer’s comments are in italic and blue colors.)

We would like to express our sincere thanks to the reviewer for his/her valuable comments and suggestions. We have revised the manuscript in accordance with the reviewer’s comments and suggestions. All the changes made in the revision are underlined. Our replies to the reviewer’s comments and suggestions are as follows.

This manuscript presents a watt-level, narrow-linewidth, tunable green external-cavity semiconductor laser (ECDL) with external-cavity synchronous-locking technique. The ECDL is composed of two green edge-emitting diode lasers (LDs), beam shaping devices and a diffraction grating. One of the LDs is locked by self-matching the spectrum of the diffraction grating, and then part of the locked light is injected into the other LD to complete synchronous locking.

The following is recommended to address properly before it can be recommended to be published.

1.Why develop Watt-level green lasers, and what are the primary application requirements they address?

Response: In special applications that require Watt-level green lasers, like laser displays, underwater communications and detection. Extremely narrow-linewidth is not necessary but high power is important, which will cause poor spatial coherence and speckle[1s,2s], as shown in Fig.2s.

References:

[1s] Akram M. Nadeem., and Xu yuan. Chen. Speckle reduction methods in laser-based picture projectors. Optical Review 23, no. 1 (2016): 108-120.

[2s] Nan Ei. Yu., Ju Won. Choi., Hee jong. Kang., and etc. Speckle noise reduction on a laser projection display via a broadband green light source. Opt. Express 22, 3547-3556 (2014).

2.In the introduction, the author mentions that “the DFB or DBR green LD is very challenging.” In what specific aspects does the challenge of the DFB or DBR green LD manifest?

Response: The following three main challenges are:1) the need for materials that can match the lattice structure to fabricate devices; 2) the requirement to be able to operate under high current density to achieve optical amplification, which is a prerequisite for laser generation; 3) and the ability to uniformly inject sufficient charge carriers into the p-type region with an imperfect band structure. The limited power (1 W) of single green LD due to the well-known “green gap”.

For the description of experimental details and measurement results, the following aspects need to be explained.

3.Is the beam profile in Fig. 1(a) measured before or after the diffraction grating?

Response: Thank you very much for pointing out the unclear points. In our experiment, beam profile in Fig. 1(a) is measured after the diffraction grating.

4.In Fig. 2, what is the operating state of the green laser during free-running? During free-running, does the feedback light from the diffraction grating return to the LD? In Fig. 2 (c), the output power of the laser free-running mode is higher than that of the other three diffraction grating states. What is the reason for the higher output power of the laser free-running mode?

Response: We admire the reviewer for this valuable comment. The green laser in freely-running regime operates in multiple Fabry–Perot modes with a wide 3dB-linewidth of 4 nm at 518 nm center wavelength. During free-running, the feedback light from the diffraction grating no return to the LD. Due to the existence of mode competition in the external-cavity feedback, laser loss occurs.

5.In the testing of the three diffraction gratings, are the tilt angles of the diffraction gratings the same, and is the laser incidence angle consistent for the different diffraction gratings?

Response: We admire the reviewer for this valuable comment. Tilt angles and the laser incidence angle is not consistent for the different diffraction gratings.

6.What’s the laser linewidth when the first LD is locked by the grating feedback? What method was used to measure the laser linewidth in Figure 3(b)? What is the current order of magnitude for the linewidth level of green lasers?

 Response: We admire the reviewer for this valuable comment. The laser linewidth when the first LD is locked by the grating feedback is 0.06nm. Laser linewidth in Figure 3(b) are measured by an optical spectrum analyzer (Ando. AQ-6315E). The current order of magnitude for the linewidth level of green lasers is pm.

For some language expressions, authors are asked to confirm.

1.The paragraphs in the text are too long. It would be better to break them into appropriate sections.

Response: Thank you for your valuable suggestions. We have tried our best to condense the paragraphs.

2.Place the table title from Table 1 above the table.

Response: We appreciate the reviewer for pointing out our mistakes. We have modified in the new revised manuscript, in the table 1.

3.Place the image and caption for Figure 4 on a single page.

 Response: We appreciate the reviewer for pointing out our mistakes. We have modified in the new revised manuscript, in the Figure 4.

Reviewer 4 Report

Comments and Suggestions for Authors

The paper presents a new green diode laser system by synchronous-locking two similar diode laser with grating in Littrow configuration, and watt-level output can be achieved. The laser system can be tuned and laser linewidth is less than 0.15 nm.

The external-cavity synchronous-locking is a novel method, and the results are good. I suggest to publish the paper in Sensors after the following comments are addressed:

1): the authors mentioned the focal length of FAC and SAC are 0.3 and 0.8 nm, and the spot size is 4.4 and 4.0 mm in the fast and slow axis direction, where was the spot size measured? With the very short focal length, the beam size is normally smaller than the measured results.

2) The beam profile measured by CCD shows two peaks, do these two peaks correspond to the two lasers? If yes, it is possible to make these two peaks overlap totally?

3) in caption for Fig. 1 a), M is a green high reflective mirror, how much is the reflectivity? Since the two laser beams are combined by this mirror, its reflectivity will determine the contribution of two lasers to the total output, and also determine the loss of the total laser system, in order to obtain a better results, how to choose the reflectivity of this mirror?

Comments on the Quality of English Language

The English could be improved to more clearly express the research.

Author Response

(Note: the reviewer’s comments are in italic and blue colors.)

We would like to express our sincere thanks to the reviewer for his/her valuable comments and suggestions. We have revised the manuscript in accordance with the reviewer’s comments and suggestions. All the changes made in the revision are underlined. Our replies to the reviewer’s comments and suggestions are as follows.

The paper presents a new green diode laser system by synchronous-locking two similar diode laser with grating in Littrow configuration, and watt-level output can be achieved. The laser system can be tuned and laser linewidth is less than 0.15 nm.

The external-cavity synchronous-locking is a novel method, and the results are good. I suggest to publish the paper in Sensors after the following comments are addressed:

1.the authors mentioned the focal length of FAC and SAC are 0.3 and 0.8 nm, and the spot size is 4.4 and 4.0 mm in the fast and slow axis direction, where was the spot size measured? With the very short focal length, the beam size is normally smaller than the measured results.

Response: We appreciate the reviewer for pointing out our ambiguity. In our experiment, the combined beam profile was detected by a visible-wavelength CCD. As you say, with the very short focal length, the beam size is normally smaller than the measured results.

2.The beam profile measured by CCD shows two peaks, do these two peaks correspond to the two lasers? If yes, it is possible to make these two peaks overlap totally?

Response: We appreciate the comments of the reviewer. Due to the inherent spot characteristics of multi-mode edge emitting semiconductor lasers, their far-field spot is an elliptical shape. In our experiment, the far-field with two peaks are caused by the spatial misalignment and the difference in the light spots of the laser diodes, which can be eliminated by further adjusting the optical path. Here, we only demonstrate a novel scheme to obtain high power and narrow linewidth laser source with compactness size. The problem of light spot can be solved by adopting single mode diode.

3.in caption for Fig. 1 a), M is a green high reflective mirror, how much is the reflectivity? Since the two laser beams are combined by this mirror, its reflectivity will determine the contribution of two lasers to the total output, and also determine the loss of the total laser system, in order to obtain a better results, how to choose the reflectivity of this mirror?

Response: We appreciate the comments of the reviewer. The reflectivity 0f M is 98.7%, in order to obtain a better results, the reflectivity of the mirror should be as high as possible to minimize losses.

Reviewer 5 Report

Comments and Suggestions for Authors

This work is undoubtedly relevant and holds significant practical interest. However, a few minor questions arose during the review:

1. Could the authors provide the optical spectra of the light reflected from the diffraction gratings? An experimental measurement of their spectral width would be of particular interest.

2. The measured linewidth of 0.06 nm appears to be very close to the resolution limit of the optical spectrum analyzer (Ando. AQ-6315E). Is a single-frequency lasing regime achievable at the threshold current?

3. Does the relevance of this work to the journal "Sensors" lie solely in the laser's potential applications? This question arises because the manuscript is dedicated to the development of a laser source, rather than a sensor system itself. It might be beneficial to elaborate on the specific areas of application in greater detail to better align the paper with the journal's scope. 

Author Response

(Note: the reviewer’s comments are in italic and blue colors.)

We would like to express our sincere thanks to the reviewer for his/her valuable comments and suggestions. We have revised the manuscript in accordance with the reviewer’s comments and suggestions. All the changes made in the revision are underlined. Our replies to the reviewer’s comments and suggestions are as follows.

This work is undoubtedly relevant and holds significant practical interest. However, a few minor questions arose during the review:

1.Could the authors provide the optical spectra of the light reflected from the diffraction gratings? An experimental measurement of their spectral width would be of particular interest.

Response: We admire the reviewer for this valuable comment. Regrettably, in our experiment, we only focused on the output spectrum but did not pay attention to the reflected spectrum. But we provided the freely spectrum of LD as well as the external-cavity spectrum, and also the grating parameters.

2.The measured linewidth of 0.06 nm appears to be very close to the resolution limit of the optical spectrum analyzer (Ando. AQ-6315E). Is a single-frequency lasing regime achievable at the threshold current?

Response: We admire the reviewer for this valuable comment. Due to the resolution limit of the optical spectrum analyzer (Ando. AQ-6315E), a single-frequency lasing regime is possibly achievable at the threshold current. Morever, as shown in Equation 1-1, when other parameters remain unchanged, the linewidth Δν of the ECDL is inversely proportional to the square of the diffraction efficiency (r3) of the external-cavity mirror and inversely proportional to the square of the external cavity l. Linewidth can be further narrowed or achieve the performance of single frequency by shortening the cavity length through PZT and increasing the grating of first-order diffraction efficiency or using etalon, etc[3s, 4s].

References:

[3s] Yu N E, Choi J W, Kang H, etc. Speckle noise reduction on a laser projection display via a broadband green light source[J]. Optics Express, 2014, 22(3): 3547-3556.

[4s] Akram M N, Chen X. Speckle reduction methods in laser-based picture projectors[J]. Optical Review, 2016, 23(1): 108-120.

3.Does the relevance of this work to the journal "Sensors" lie solely in the laser's potential applications? This question arises because the manuscript is dedicated to the development of a laser source, rather than a sensor system itself. It might be beneficial to elaborate on the specific areas of application in greater detail to better align the paper with the journal's scope. 

Response: We admire the reviewer for this valuable comment. Laser technology can be utilized for laser sensing, which enables non-contact and long-distance measurement. It has the advantages of high speed, high precision, wide range, and strong resistance to photoelectric interference. It has been widely applied in fields such as national defense, production, medicine, and non-electrical measurement.

Round 2

Reviewer 1 Report

Comments and Suggestions for Authors

The authors have revised the manuscript and provided responses in accordance with the reviewers' comments.

Reviewer 3 Report

Comments and Suggestions for Authors

The authors have answered the question properly in the revised paper . I would like to recommend it to be published on Sensors.